# COVID-19 symptom load as a risk factor for chronic pain: A national cross-sectional study

**Jamie L. Romeiser**[1]*, **Christopher P. Morley**[1], **Sunitha M. Singh**[2,3]

**1** Department of Public Health and Preventive Medicine, Upstate Medical University, Syracuse, New York, United States of America, **2** Department of Perioperative Surgical Services, Stony Brook University Medical Center, Stony Brook, New York, United States of America, **3** Department of Anesthesiology, Stony Brook University Medical Center, Stony Brook, New York, United States of America

* RomeiseJ@upstate.edu

## Abstract

### Introduction

Emerging evidence suggests that a COVID-19 infection with a high initial severity may be associated with development of long-COVID conditions such as chronic pain. At the population level, it is unknown if severity of a COVID-19 infection might be a new risk factor for chronic pain above and beyond the traditional slate of pre-established risk factors. The purpose of this study is to examine whether COVID-19 severity of infection may be a new risk factor for chronic pain.

### Methods

Using data from the 2021 National Health Interview Survey (n = 15,335), this study examined the adjusted odds of experiencing high frequency levels of pain in the past 3 months for those who reported no/mild symptoms from a COVID-19 infection, and those reporting moderate/severe symptoms from COVID-19, compared to those never infected. A 1:1:1 propensity score matched analysis was also performed to examine the odds of pain.

### Results

Prevalence of pain was higher in the moderate/severe symptom group compared to the no infection group (25.48% vs 19.44%, p <0.001). Both the adjusted model (odds ratio [OR] = 1.28, 95% confidence interval [CI] = 1.09, 1.51) and matched model (OR = 1.45, CI = 1.14, 1.83) revealed higher odds of pain for those with moderate/high COVID-19 symptoms compared to no infection.

### Conclusions

A moderate/highly symptomatic COVID-19 infection may be a new risk factor for chronic pain. As the absolute number of severe COVID-19 infections continues to rise, overall prevalence of chronic pain may also increase. While knowledge continues to unfold on long-haul symptoms, prevention of severe infections remains essential.

**Data Availability Statement:** All 2021 National Health Interview Survey (NHIS) are publicly available and may be found here: https://www.cdc.gov/nchs/nhis/2021nhis.htm.

**Funding:** The author(s) received no specific funding for this work.

**Competing interests:** The authors have declared that no competing interests exist.

## Introduction

The long-term effects of COVID-19 on the body is a growing public health issue. As of March 2023, approximately 760 million confirmed cases coronavirus disease 2019 (COVID-19) have been documented globally, including 6.87 million deaths [1]. Survivors of COVID-19 may experience what has been referred to as a 'constellation' of lingering symptoms or conditions even after clearance of the virus. These symptoms and conditions can present for different lengths of time. Symptoms that persist longer than three months, and that did not exist prior to contracting the virus, are known as long-COVID symptoms [2].

The prevalence of long-COVID may be underestimated. In June 2022, the U.S. Census Bureau and the National Center for Health Statistics estimated that 19% of those who had COVID-19 in the past also had lingering symptoms, with 7.5% of all adults in the US reporting long-COVID symptoms [3]. However, a recent meta-analysis found that 45% of COVID-19 survivors reported at least one lingering symptom [4]. While there is still debate amongst the literature to define significant and reliable predictors of long-COVID [5], hospitalization [6] and the presence of a more severe initial symptom load [7] may be linked to persistence of residual symptoms. Notably, amongst separate hospitalized and non-hospitalized population analyses, one of the top five reported long-COVID symptoms was pain [4].

The exact underlying biological mechanism(s) of long-COVID pain and painful manifestations remains unclear, but it could include direct and indirect damage by the virus [8]. Long-COVID chronic pain has been linked to severity of initial symptom load [2, 9]. With lingering chronic pain as prevalent long-COVID symptom, then it's conceivable to think that COVID-infection and symptom load may arise as a new determinant of chronic pain.

Chronic pain is a significant public health issue [10]. Approximately 20% of adults suffer from chronic pain [11, 12], which is widely defined as frequent pain that lasts for at least three months [12–15]. Socioeconomic status and education levels are often inversely associated with pain [16, 17], whereas age and body mass index are positively associated with pain frequency [13, 16, 17]. Race, gender, and presence of comorbid conditions such as cancer, inflammatory diseases, and diabetes, are commonly linked to chronic pain in population-based studies [16, 17]. With widespread history of COVID-19 infection following a years-long pandemic, the addition of COVID-19 as a new predictor of chronic pain would be a significant contribution to the slate of risk factors and moderators that are already known. Further, the impact of symptom severity on the likelihood of later reported pain would be a useful predictor of long-COVID in patients.

To our knowledge, there are no population-based studies that examine the role of severity of COVID-19 infection as a determinant of chronic pain, above and beyond the traditional risk factors for chronic pain. Using the National Health Interview Survey (NHIS) 2021 data, the aim of this study is to investigate the association between a COVID-19 diagnosis and severity of initial symptoms, and self-reported daily pain.

## Methods

This study was conducted using data from the 2021 National Health Interview Survey (NHIS, n = 29,482) [18]. The NHIS is a nationally representative survey of the non-institutionalized United States population. Detailed information on the survey design, implementation, data collection, application of survey weights, and access to the data may be found here: https://www.cdc.gov/nchs/nhis/index.htm. Briefly, this annual, cross-sectional household survey was

performed by the National Center for Health Statistics (NCHS), which is part of the Centers for Disease Control and Prevention (CDC). Geographic cluster sampling techniques are used to select participants 18 years of age and older for face-to-face interviews. Multi-level sampling methods are deployed to ensure the each month of data collection is representative of the United States populations [18]. Survey data collection protocols are approved by the NCHS Ethics Review board as a public health surveillance activity (#2019–09 National Health Interview Survey). Data are deidentified and made publicly available, therefore, University specific IRB review is exempt for this study.

### Inclusion criteria

Participants included in this study were at least 18 years of age. Individuals who did not have pain data, and individuals who reported never taking a COVID-19 test in the past were excluded. Because of the potential to confound the results, individuals who were involved in an accident in the past 3 months were excluded from the analysis.

### Primary outcome

The primary outcome of this study was frequency of pain, assessed in the NHIS survey as: "In the past three months, how often did you have pain?" Responses were dichotomized as never having pain or reporting some days with pain, and pain on most days or every day. This classification is consistent with prior literature to indicate those suffering from chronic pain [12–14].

### Primary predictor

Individuals were classified into three groups based on COVID-19 testing status and symptom status. Individuals who had reported having taken a COVID-19 test in the past, testing negative, and never having a diagnosis of COVID-19 by a medical professional were considered to be COVID-19 negative. Those who had reported ever taking a COVID-19 test and testing positive were additionally asked to report the worst level of their COVID-19 related symptoms. These participants were then divided into symptom load groups: those who reported being asymptomatic or reported mild symptoms, and those who reported moderate or severe symptoms. Sensitivity analyses were conducted to ensure these groupings were appropriate. This was done by first examining the distribution in covariates and pain scores for participants in the asymptomatic group versus mild symptoms group. These two groups were found to be similar to one another, which would indicate that neither group alone would drive the analysis results. We also examined the distribution in covariates and pain scores for participants in the moderate symptoms group versus severe symptoms groups, and found similar results, thereby confirming the appropriateness of the symptomatic groups.

### Covariates

Established predictors of pain, and predictors of both pain and early pandemic diagnosis of COVID-19 were identified [5–7, 9, 13, 16, 17]. Socio-demographic variables included age (recategorized into decades), sex (male, female), race/ethnicity (non-Hispanic White, non-Hispanic Black, Asian, Hispanic, other single or multiple races), education (high school graduate or less, some college but no degree, academic or vocational Associates degree, Bachelor's degree, higher degree), BMI (underweight, normal weight, overweight, obese), and poverty income ratio (defined as the ratio of family income to poverty threshold, which was further classified to around or below the poverty line [PIR 0–1.24], medium income [PIR 1.25–2.99]

and high income [PIR 3+]). Chronic or comorbid variables included weak immune system due to health conditions, prior diagnosis of diabetes, self-reported difficulty walking, and prior diagnosis of chronic inflammation (arthritis, rheumatoid arthritis, gout, lupus, or fibromyalgia).

### Statistical analysis

Complex survey weights were applied to all analyses to account for the complex survey design. Categorical variables are reported as unweighted frequencies and percents within Table 1. Weighted proportions (i.e., the application of the complex survey weights) are reported both in the tables and described within the narrative. Missing data were minimal (3.3%), and therefore not imputed. Complex survey weights were used for all inferential analyses. Bivariate associations with pain were assessed using chi-square tests and logistic regression for main predictor and all covariates. All variables were found to be significantly associated with pain, and included in an adjusted multiple logistic regression model.

To further examine the association between COVID-19 symptom load and pain, a 1:1:1 propensity score matched analysis was performed. In brief, a propensity score is the probability of receiving a treatment/exposure conditioned upon a set of observable characteristics [19]. Propensity score matching is a technique used to balance uneven distributions of covariates across treatment groups in observational data [20]. The goal is to obtain treatment/exposure groups with no significant discernable differences in observable covariates between the primary exposure groups, similar to what is achieved through the process of randomization [21, 22]. In this case, there was a significant imbalance of covariates among the three COVID-19 groups, and propensity score matching was implemented to decrease potential confounding. In this study, a three-way matched sample was chosen using an overlapping pairwise approach [23]. Each COVID-19 group contrast was examined for possible overlapping matches (i.e., COVID-19 negative vs low symptom load, COVID-19 negative vs high symptom load, and low symptom versus high symptom). Propensity score matching models were constructed using all covariates from the adjusted model. Matching was performed using the PSMATCH procedure in SAS software, with a greedy-nearest neighbor approach without replacement, and a caliper width of 0.15 the standard deviation of the logit of the propensity score. Original survey weights were preserved [24]. A final covariate balance assessment was performed to ensure covariates were balanced amongst all three matched COVID-19 groups (S1 Table). All group and variable level contrasts demonstrated an absolute standardized mean difference of <0.1, indicating negligible differences in covariates. Upon demonstration of covariate balance, a final weighted logistic regression model was performed to examine the association between COVID-19 status/symptom load and pain in the matched data. All analyses were performed using SAS © 9.4 software (Cary, NC).

### Results

A total of 15,335 individuals met the inclusion criteria, representing approximately 135.7 million US adult citizens. A total of 12,131 reported never testing positive for COVID-19 (76.74%), 1440 reported testing positive with mild/no symptoms (10.66%), and 1764 reported testing positive with moderate or severe symptoms (12.60%) (Table 1). Those reporting no/ infrequent pain encompassed 80.34% of the sample (n = 11,965), with 19.66% reporting frequent (chronic) pain (n = 3370).

In unadjusted analyses, there was a significant difference in prevalence of frequent pain by COVID-19 symptom group (Chi-Square p <0.001). Of those who never contracted COVID-19, 19.44% reported frequent pain. Only 14.36% of who were asymptomatic or had mild

**Table 1. Demographic and clinical characteristics.**

| Characteristics | Unweighted | | Weighted |
|---|---|---|---|
| | **N** | **%** | **%** |
| **COVID-19** | | | |
| No Infection | 12131 | 79.11% | 76.74% |
| COVID+, Asymptomatic/Mild | 1440 | 9.39% | 10.66% |
| COVID+, Moderate/Severe | 1764 | 11.50% | 12.60% |
| **Age Decades** | | | |
| 18–29 | 2327 | 15.21% | 22.53% |
| 30–39 | 2733 | 17.87% | 18.08% |
| 40–49 | 2444 | 15.98% | 16.76% |
| 50–59 | 2506 | 16.38% | 16.03% |
| 60–69 | 2611 | 17.07% | 14.20% |
| 70–79 | 1868 | 12.21% | 8.88% |
| 80+ | 808 | 5.28% | 3.62% |
| **Sex** | | | |
| Female | 8585 | 55.99% | 53.34% |
| Male | 6749 | 44.01% | 46.66% |
| **Race Ethnicity** | | | |
| Non-Hispanic White | 9837 | 64.15% | 60.53% |
| Non-Hispanic Black | 1824 | 11.89% | 12.75% |
| Hispanic | 2377 | 15.50% | 18.67% |
| Asian | 912 | 5.95% | 6.00% |
| Other single or multiple races | 385 | 2.51% | 2.46% |
| **Education** | | | |
| HS Graduate or less | 4639 | 30.39% | 34.77% |
| Some College, no degree | 2309 | 15.13% | 15.41% |
| Associates (academic or vocational) | 1887 | 12.36% | 11.39% |
| Bachelor's Degree | 3885 | 25.45% | 24.03% |
| Higher Degree | 2543 | 16.66% | 14.41% |
| **Poverty Income Ratio** | | | |
| 0–1.24 (low) | 1973 | 12.87% | 13.27% |
| 1.25–2.99 (middle income) | 4274 | 27.87% | 28.47% |
| 3.0–5 (high income) | 9088 | 59.26% | 58.26% |
| **BMI** | | | |
| Underweight | 226 | 1.51% | 1.72% |
| Healthy weight | 4702 | 31.38% | 31.10% |
| Over weight | 5150 | 34.37% | 33.92% |
| Obese | 4905 | 32.73% | 33.26% |
| **Chronic Inflammation Diagnosis** | 3665 | 23.92% | 19.64% |
| **Weakened Immune System** | 812 | 5.31% | 4.79% |
| **Diabetes** | 1506 | 9.83% | 8.96% |
| **Difficulty Walking / Functional Limitations** | | | |
| No/Some | 14554 | 94.93% | 96.02% |
| A lot/Cannot walk | 778 | 5.07% | 3.98% |
| **Presence of Chronic Pain** | | | |
| No | 11965 | 78.02% | 80.34% |
| Yes | 3370 | 21.98% | 19.66% |

HS = High School; BMI = Body Mass Index

symptoms reported frequent pain, whereas 25.48% of those with moderate/severe symptoms reported frequent pain.

## Adjusted regression

All covariates were also independently associated with pain [Table 2]. After controlling for age, sex, race/ethnicity, education, BMI, poverty income ratio, compromised immune system, diabetes, self-reported difficulty walking, and chronic inflammation in a multiple logistic regression model–those who had moderate or severe symptoms during their COVID-19 infection were 1.28 times more likely to report being in pain most/every day in the past 3 months compared to the no infection group (OR: 1.28, CI: 1.09, 1.51) (Table 2). Interestingly, those who were asymptomatic or reported mild symptoms were less likely to report pain in the past 3 months compared to the no infection group (OR: 0.81, CI: 0.69, 0.96). Adjusted probabilities are presented in Fig 1A, demonstrating that the adjusted probability of chronic pain was approximately 4 percentage points higher amongst those who had a higher COVID-19 symptom load compared to those who had not contracted COVID-19 (20% vs. 16%, respectively).

## Propensity score matched analysis

There were significant imbalances of covariates within the COVID-19 symptom groups (S1 Table). For example, as education levels increased, the proportion of those who did not contract COVID-19 also increased. A similar relationship was found with poverty income ratio, with those in the higher income group having a lower likelihood of contracting COVID-19. Those in the lower symptom group appeared to have fewer comorbidities and were younger compared to those who never contracted COVID-19. Many of the established risk factors of pain were also associated with COVID-19 infection. Therefore, a pairwise propensity score matching process was used to select matched participants within the three COVID-19 groups, matching on all covariates listed above. After propensity score matching, all three groups were balanced in terms of their distribution of covariates (S1 Table). Otherwise said, after matching, there were no significant differences between the three matched COVID-19 groups in terms of age, sex, race, education, BMI, PIR, and additional comorbidities. Each COVID-19 group contained 1223 participants, for a total matched sample size of 3669 participants.

After matching, the odds of pain in the moderate/severe group strengthened slightly compared to the adjusted model (OR: 1.45 [CI: 1.14, 1.83], p = 0.002) (Table 3), and the predicted probability of pain was approximately 6 percentage points higher amongst the high symptom group (22% vs 16%, Fig 1B). The odds of pain for the mild/asymptomatic group became insignificant compared to the no infection group (0.86 [CI: 0.68, 1.09], p = 0.20).

## Discussion

As the number of people who contract COVID-19 continues to rise, it becomes increasingly important to quantify the burden of long-term effects on the population. In our study, those who experienced a higher COVID-19 symptom burden also had a higher prevalence of chronic pain. At the population level, even after adjusting for a multitude of risk factors, COVID-19 severity was significantly associated with chronic pain. The adjusted predicted probability of pain rose from 16% in those who never tested positive for COVID-19, to 20% in those with a high COVID-19 symptom burden.

Chronic pain is a significant public health issue [10, 25]. Global prevalence estimates of adults who suffer from chronic pain can range, but likely fall around 20% [11]. This estimate is consistent with our sample wherein the overall prevalence of pain (on most days or every day) was approximately 20%. Our findings contrasted another study conducted using 2020 NHIS

**Table 2. Unadjusted and adjusted (weighted) logistic regression predicting chronic pain.**

| Predictors of Chronic Pain | Unadjusted Odds Ratio (95% CI) | p-value | Adjusted Odds Ratio (95% CI) | p-value |
|---|---|---|---|---|
| **COVID-19** | | | | |
| No Infection | *Reference* | | *Reference* | |
| COVID+, Asymptomatic/Mild | 0.70 (0.59, 0.81) | <0.0001 | 0.81 (0.69, 0.96) | 0.02 |
| COVID+, Moderate/Severe | 1.42 (1.24, 1.62) | <0.0001 | 1.28 (1.09, 1.51) | 0.002 |
| **Age Decades** | | | | |
| 18–29 | *Reference* | | *Reference* | |
| 30–39 | 1.62 (1.32, 1.99) | <0.0001 | 1.45 (1.17, 1.79) | <0.0001 |
| 40–49 | 2.60 (2.12, 3.20) | <0.0001 | 2.04 (1.64, 2.54) | <0.0001 |
| 50–59 | 4.07 (3.36, 4.92) | <0.0001 | 2.30 (1.87, 2.83) | <0.0001 |
| 60–69 | 4.48 (3.71, 5.39) | <0.0001 | 1.93 (1.56, 2.38) | <0.0001 |
| 70–79 | 4.92 (4.03, 6.00) | <0.0001 | 1.70 (1.36, 2.13) | <0.0001 |
| 80+ | 5.41 (4.25, 6.89) | <0.0001 | 1.53 (1.14, 2.05) | 0.005 |
| **Sex** | | | | |
| Female | 1.14 (1.04, 1.24) | <0.01 | 1.00 (0.91, 1.13) | 0.85 |
| Male | *Reference* | | *Reference* | |
| **Race Ethnicity** | | | | |
| Non-Hispanic White | *Reference* | | *Reference* | |
| Non-Hispanic Black | 0.75 (0.65, 0.87) | <0.0001 | 0.61 (0.51, 0.72) | <0.0001 |
| Hispanic | 0.61 (0.53, 0.70) | <0.0001 | 0.61 (0.51, 0.72) | <0.0001 |
| Asian | 0.27 (0.21, 0.36) | <0.001 | 0.40 (0.29, 0.54) | <0.0001 |
| Other single or multiple races | 1.04 (0.77, 1.40) | 0.81 | 1.14 (0.80, 1.63) | 0.46 |
| **Education** | | | | |
| HS Graduate or less | 1.79 (1.55, 2.06) | <0.0001 | 1.37 (1.15, 1.64) | <0.001 |
| Some College, no degree | 1.69 (1.44, 1.99) | <0.0001 | 1.58 (1.30, 1.91) | <0.0001 |
| Associates (academic or vocational) | 2.28 (1.93, 2.69) | <0.0001 | 1.90 (1.57, 2.30) | <0.0001 |
| Bachelor's Degree | 1.14 (0.98, 1.32) | 0.09 | 1.17 (0.99, 1.38) | 0.07 |
| Higher Degree | *Reference* | | *Reference* | |
| **Poverty Income Ratio** | | | | |
| 0–1.24 (low) | *Reference* | | *Reference* | |
| 1.25–2.99 (middle income) | 0.79 (0.68, 0.91) | <0.01 | 0.77 (0.64, 0.91) | 0.003 |
| 3.0–5 (high income) | 0.56 (0.49, 0.65) | <0.0001 | 0.60 (0.50, 0.71) | <0.0001 |
| **Body Mass Index** | | | | |
| Underweight | 1.20 (0.79, 1.82) | 0.38 | 1.12 (0.69, 1.83) | 0.65 |
| Healthy weight | *Reference* | | *Reference* | |
| Over weight | 1.36 (1.20, 1.54) | <0.0001 | 1.09 (0.95, 1.25) | 0.24 |
| Obese | 2.17 (1.92, 2.45) | <0.0001 | 1.38 (1.20, 1.59) | <0.0001 |
| **Arthritis Diagnosis** | 6.99 (6.32, 7.74) | <0.0001 | 4.32 (3.82, 4.88) | <0.0001 |
| **Weakened Immune System** | 3.72 (3.11, 4.44) | <0.0001 | 1.95 (1.58, 2.40) | <0.0001 |
| **Diabetes** | 2.61 (2.28, 2.99) | <0.0001 | 1.25 (1.05, 1.47) | 0.01 |
| **Difficulty Walking / Functional Limitations** | | | | |
| No/Some | *Reference* | | *Reference* | |
| A lot/Cannot walk | 12.37 (10.22, 14.97) | <0.0001 | 5.03 (3.98, 6.35) | <0.0001 |

Adjusted model C-Index = 0.78. CI = Confidence Interval; HS = High School

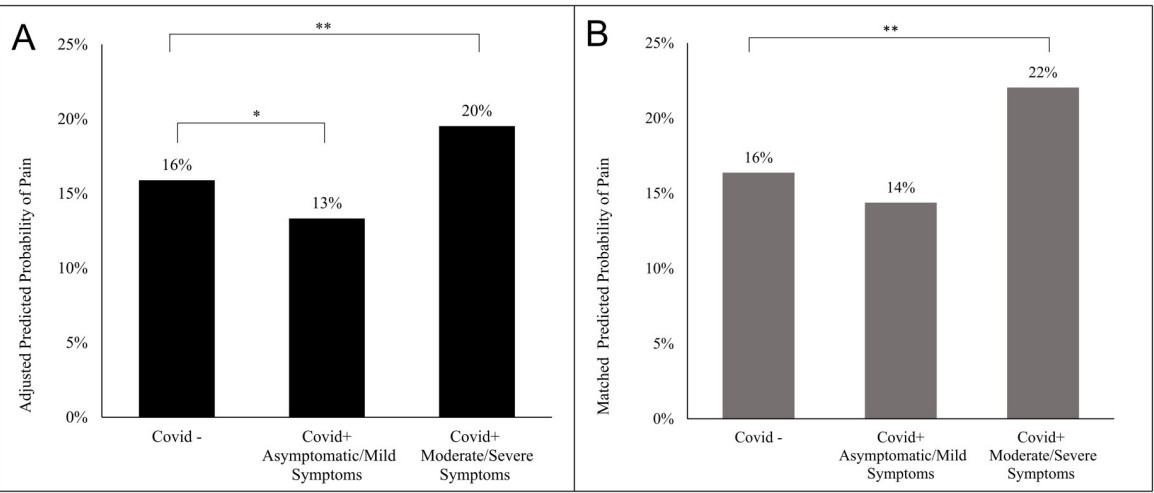

**Fig 1. Predicted probabilities of pain by COVID-19 symptom group.** After adjusting for multiple covariates, COVID-19 symptom group was found to be significantly associated with pain. While those with moderate/severe symptoms had a higher probability of pain, those who were asymptomatic/mildly symptomatic had a lower probability of pain compared to those who were never infected (Panel A). After matched pairing on multiple covariates, only those in the higher symptomatic group had a significantly different (higher) probability of pain compared to those never infected (Panel B). * Indicates p-value = 0.02; ** Indicates p-value of <0.01.

data [7]. This study similarly examined the outcome of pain for different COVID-19 symptomatic loads, but respondents were over 65, and results were not adjusted for covariates. In our study, as with prior studies, socioeconomic status, race, age, education, body mass index, and comorbid conditions were also associated with chronic pain and COVID-19 status. Therefore, adjustment was necessary to reduce confounding of the primary hypothesis.

Compared to those who tested negative, those who tested positive for COVID-19 were more likely to be non-white, of working age, with lower income and education levels, and a higher of comorbid conditions. This was similar to other studies conducted during this time period [26]. Therefore, an additional 1:1:1 matching approach was chosen to balance these covariates and further isolate and assess the potential impact of COVID-19 on chronic pain. Matching was successfully achieved for most participants in the COVID-19 group with the lowest frequency (i.e., 1223/1440 of mild symptom group, or 85%). After matching, the odds of pain in the high symptom group increased and remained significant. There was no difference in the likelihood of pain between the COVID-19 negative group and the no/mild symptom group. This might be explained by the socio-demographic characteristics in the no/mild symptom group. These individuals were younger, with a lower proportion of functional disability and historic diagnosis of chronic inflammation (S1 Table). These findings are similar to other studies, where younger and healthier individuals were at lower risk of a severe COVID-

**Table 3. Matched logistic regression predicting pain.**

| COVID-19 Group | Odds Ratio (95% CI) | P-Value |
|---|---|---|
| No Infection | *Reference* | |
| COVID+, Asymptomatic/Mild | 0.86 (0.68, 1.09) | 0.20 |
| COVID+, Moderate/Severe | 1.45 (1.14, 1.83) | 0.002 |

CI = Confidence Interval

19 illness and poor outcomes [27, 28]. Considering these factors are also strongly associated with pain, this may explain why after matching the three COVID-19 groups, the protective association seen in the adjusted model was no longer significant. It's likely there are additional confounders that were unmeasured or unaccounted for that might further explain why those in the mild symptom group seemed to have a lower prevalence of chronic pain.

While the present study does not provide evidence of a causal relationship between COVID-19 symptom load and development of chronic pain due to the cross-sectional nature of the data, there are a handful of studies have examined newly onset chronic pain after a COVID-19. One case-controlled cross-sectional study of 119 patients in Brazil compared newly onset chronic pain amongst hospitalized patients with and without a COVID-19 infection [29]. Results showed that while pre-hospitalization pain was much higher in the control group, de novo chronic pain was more prevalent in the COVID-19 infection group (19.6%) compared to the control group (1.4%). Another cross-sectional study from Cyprus included 90 COVID-19 survivors [30]. The prevalence of chronic pain was found to be around 63%, while newly onset chronic pain occurred in one in six participants, or 16.7%. Finally, a large multi-center trial of patients previously hospitalized from a COIVD-19 infection was conducted in Spain [31]. Prevalence of musculoskeletal pain was 45%, and 22.6% reported developing newly onset chronic pain.

De novo pain appears to be occurring, especially after severe infections. Therefore, it's plausible that overall estimates of the prevalence of pain in the population might also increase. In the U.S., studies using the NHIS survey data from 2016–2019 have reported the prevalence of chronic pain to be relatively steady at 20.4% [13, 14]. In this 2021 survey data however, the prevalence of reported chronic pain rose to 20.9% in the full study population. Pain prevalence in the full population database was slightly higher compared to our inclusion sample (20.9% vs 19.7%). Our study inclusion criteria were partially based on taking a COVID-19 test. Therefore, this difference in prevalence of pain may be due to differences in those who had access to COVID-19 testing. It's unknown to what degree this might affect the model results. In general, it's possible that our sample may be underestimating the associated impact of a severe COVID-19 infection on pain due to lack of reported COVID-19 testing at the time.

The magnitude of the public health burden of pain has proven difficult to quantify [32]. Chronic pain is associated with an increase in depression, anxiety, and sleep disturbance [33]. It can impact daily living activities, reduce social engagement, and impact the ability to contribute to the workforce [12]. The economic impact of chronic pain is astonishing. The cost attributable to chronic pain through medical expenditures and loss of worker productivity in the United States is estimated to range from 560 to 635 billion dollars per year [34].

Further, treating chronic pain is complex. Nonpharmacologic approaches such as physical therapy, massage therapy, chiropractic care, or even forms of music therapy can serve as viable alternatives to pharmacologic treatments for pain relief. However, these services are still underutilized [35]. Limited insurance coverage and disparities in utilization of these services for pain management and post COVID-19 symptom management have been documented [36]. Much of pain management still revolves around an 'opioid-centric' approach [37]. In fact, a recent study found an increase in opioid prescribing patterns for treating post covid-19 symptoms [8]. These findings were described in the population of Veterans Health Administration (VHA) users, but the implication of using opioids to manage pain, and the subsequent impact on the current opioid epidemic in the general population, are similar. Opioid prescription rates were plateauing and declining in years prior to the pandemic [38], but remained alarmingly high in some areas of North America. It is currently unknown to what degree these rates have changed throughout the pandemic. Notwithstanding, opioid prescription related

mortality rates rose significantly in 19 states from 2019 to 2020, and were involved in nearly a quarter of opioid-related mortalities in 2020 [39].

This study has implications for public health practice and policy. Several characteristics are linked to developing severe COVID-19 infections. It remains critically important to educate individuals and the public on assessing personal risk of developing a severe infection, and how to mitigate these risks. Vaccination remains one of the strongest defenses against development of a severe COVID-19 infection [40] and subsequent long-haul symptoms such as chronic pain. Vaccine efficacy remains higher against severe infections compared to milder infections, but still can wane over time against emerging variants [40]. Therefore, booster dosing programs, advocacy, and educational campaigns remain as essential components of prevention. If prevalence of chronic pain is expected to increase in the post pandemic era, then strategies for pain management must intersect at the clinical practice level and population health level [41].

Our study also has several strengths. To our knowledge, this is the first population-level multivariable-controlled and matched study to examine symptomatic COVID-19 infection as a new risk factor for chronic pain. We used the National Health Interview Survey, which in 2022 was evaluated to be the best US population level surveillance source for monitoring pain [42]. The large sample size allowed us to control for many socio-demographic factors as well as comorbid conditions that are associated with pain.

Our study also has limitations. First, as with any cross-sectional data, one cannot establish time ordered events or causality. It is possible chronic pain occurred in participants prior to a COVID-19 infection. Second, survey data are self-reported and subject to recall bias. This data did not include information on COVID-19 vaccination status or additional COVID-19 related treatments, so we were unable to account these factors. Also, it is likely that our analysis excluded some important risk factors for pain. While some behavioral risk factors were not included in the database, at the suggestion of a reviewer, we ran a model that included both smoking and hypertension as additional covariates. However, this did not appreciably improve model fit or change the effects for the primary predictor in any meaningful way, therefore, we elected to maintain the more simplified original model. Finally, to obtain the most robust identification of a COVID-19 infection, we included only those survey participants who reported taking at least one COVID-19 test. This excluded approximately 40% of the survey respondents. There may have been different propensities to take a COVID-19 test, stigmas surrounding COVID-19 testing and infection, and inequalities in testing availability [43]; all which may explain the lack of testing amongst survey respondents. Aside from additional characteristics that may be associated with non-testing, those who are infected but asymptomatic are less likely to take a test. Therefore, it's likely COVID-19 was underdiagnosed during this time.

## Conclusions

Using nationally representative population-level data, this study suggests that a highly symptomatic COVID-19 infection may be a new significant risk factor for chronic pain. This finding may foreshadow an increase in the population prevalence of chronic pain, and highlights the continued importance of reducing severe COVID-19 infections. Future prospective studies are necessary to assess the risk of chronic pain incidence after a severe COVID-19 infection.

## Supporting information

**S1 Table. Covariate distribution amongst COVID-19 groups before and after matching.** Covariates distributions were compared using Chi-Square tests, and most were found to be imbalanced amongst the COVID-19 groups prior to matching. After matching, no significant differences were found, and all standardized mean differences were <0.1, indicating good

balance.
(DOCX)

## Author Contributions

**Conceptualization:** Jamie L. Romeiser, Sunitha M. Singh.

**Data curation:** Jamie L. Romeiser.

**Formal analysis:** Jamie L. Romeiser.

**Investigation:** Jamie L. Romeiser, Sunitha M. Singh.

**Methodology:** Jamie L. Romeiser, Sunitha M. Singh.

**Project administration:** Jamie L. Romeiser.

**Resources:** Jamie L. Romeiser, Christopher P. Morley, Sunitha M. Singh.

**Software:** Jamie L. Romeiser.

**Supervision:** Jamie L. Romeiser.

**Validation:** Jamie L. Romeiser.

**Visualization:** Jamie L. Romeiser.

**Writing – original draft:** Jamie L. Romeiser, Christopher P. Morley, Sunitha M. Singh.

**Writing – review & editing:** Jamie L. Romeiser, Christopher P. Morley, Sunitha M. Singh.

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
