## [Decision Letter · Decision Letter 0]

24 Apr 2023

PONE-D-23-09591COVID-19 Symptom Load as a Risk Factor for Chronic Pain: A National Cross-Sectional StudyPLOS ONE

Dear Dr. Romeiser,

Thank you for submitting your manuscript to PLOS ONE. After careful consideration, we feel that it has merit but does not fully meet PLOS ONE’s publication criteria as it currently stands. Therefore, we invite you to submit a revised version of the manuscript that addresses the points raised during the review process.

We look forward to receiving your revised manuscript.

Kind regards,

Dong Keon Yon, MD, FACAAI

Academic Editor

PLOS ONE

Journal Requirements:

Reviewers' comments:

Reviewer's Responses to Questions

**Comments to the Author**

1. Is the manuscript technically sound, and do the data support the conclusions?

Reviewer #1: Yes

Reviewer #2: Yes

Reviewer #3: Yes

2. Has the statistical analysis been performed appropriately and rigorously? 

Reviewer #1: Yes

Reviewer #2: Yes

Reviewer #3: Yes

3. Have the authors made all data underlying the findings in their manuscript fully available?

Reviewer #1: Yes

Reviewer #2: Yes

Reviewer #3: Yes

4. Is the manuscript presented in an intelligible fashion and written in standard English?

Reviewer #1: Yes

Reviewer #2: Yes

Reviewer #3: Yes

5. Review Comments to the Author

Reviewer #1: It is no doubt that the author included a huge data and this will increase the reliability of the results but I wonder why he ignored some an important risk factors such as smoking, drinking, and hypertension. Anyhow I think this manuscript is eligible for publishing as it.

Reviewer #2: PLOS One:

Title: COVID-19 Symptom Load as a Risk Factor for Chronic Pain: A National Cross-Sectional Study.

Comments to the authors:

- There is a fundamental question the MS did not answered anywhere: how the authors confirmed the severity of COVID-19 severity?

- What are the traditional risk factors for chronic pain? This should be elaborated

- The authors made a great effort in statistical analysis to control the possible confounders which is a novel approach for this MS.

Introduction:

In this section the authors did not define the term “chronic pain” and its significance.

Line 44: 19% of those instead of just 19% those?

Methods:

- Line 81: URL for this data should be referenced.

- What do you mean by valid pain data? How did you validate the presence of chronic pain among the survey population?

- The classification of severity (apart from tests results) was not clear which symptoms (symptoms load) the authors identified to classify their participants accordingly?

- What are the variables used to conduct sensitivity analyses (and how many analyses were required) and what were the possible scenarios evolved?

- How the poverty income ratio was calculated? What was its significance?

- Multi-survey weights were applied and complex survey weights were used: what are the differences?

- Line 117: remove “simple” and replace it with bivariate logistic regression.

- I think in this section you have to mention the sensitivity analyses followed by propensity score and matching: for each you have to mention the importance and applicability to your research.

- Line 126 to line 127: this paragraph is confusing and needs more elaboration.

Results:

- How many were included in the original survey?

- Remove representing 135.7 million US adult citizens.

- In table 1: you have to mention “the weightage” unweighted and weighted?????

- Table 1: the title of the table is not full: table should be stand alone.

- Table 1: use effective digits if possible: 79.11% (to 79.1%)

- Line 151: mention the Odds ratio and C.I for the prevalence of chronic pain among the different groups.

- Table 1: bivariate analysis should be included

- Table 2: the same comments regarding the adjustment

- Please revise the Odds ratio for gender (1.14 “1.04-1.24’)????

- Figure 1A: the probability of chronic pain (not just pain is 4% higher)!!! In this figure, the dose-relationship is not working for COVID-19, as the gradient for pain is less in mild vs. COVID-19 negative, this needs explanation????

- Line 171: COVID-19 symptoms group was found to be significantly associated with pain???? Needs elaboration: how was significantly associated???

- Line 179 to line 187: this section should be placed under the statistical analysis section

Discussion section:

- Line 224: confusing “in the lowest frequency group”????

- A separate section should be dedicated to study’s limitation “most important of which is the built-in problem of cross-sectional design and scarcity of evidence to support your conclusion.

- Provide some explanation of your findings: the conflicting findings of pain in relation to mild vs. negative cases (for example).

- The conclusion section should replace that in the abstract (more realistic)

Reviewer #3: Dear Editor and Authors,

Thank you for the opportunity to review this manuscript.

It is a cross-sectional population-based study assessing whether COVID-19 symptom severity could be a risk factor for self-reported chronic pain. Data were obtained from the National Health Interview Survey (NHIS) for the year 2021. Authors categorised the primary predictor based on testing status and symptom load into three categories: no-infection, asymptomatic/mild and moderate/severe symptoms. The primary outcome was dichotomised into never having pain and reported pain groups. A complex survey design and multi-level survey weights were used. Multivariate logistic regression analysis and 1:1:1 propensity score matching was used to account for all covariates and ensure balance and better estimates, which was feasible with the large sample size of 15,335 US individuals. A total of 3669 participants were matched.

Authors found that the prevalence of chronic pain was higher in the moderate/severe symptoms group (25.48%) than in the reference group (19.44%). Adjusted regression analysis revealed that individuals in the moderate/severe symptoms group had significantly higher odds of chronic pain (OR: 1.28, 95%CI: 1.09 to 1.51) than those with no infection. Remarkably those in the asymptomatic/mild symptoms groups had lesser odds of chronic pain (OR: 0.81, 95%CI: 0.69 to 0.96) than the reference group.

Similarly, matched adjusted regression model showed significantly higher odds of pain for the moderate/severe group than the reference group (OR: 1.45, 95%CI: 1.14 to 1.83). However, matching revealed no significant difference between the mild/ asymptomatic and the no-infection group (OR: 0.86, 95%CI: 0.68 to 1.09). Adjusted probability of chronic pain was also higher in those with moderate/severe symptoms than in the reference and the asymptomatic/mild symptoms groups, 20%, 16%, and 13%, respectively. Matched paired probabilities were also similar (22%, 16%, and 14%).

My overall impression is positive; despite the cross-sectional nature limitations, methods and analysis were rigorous, and key limitations were highlighted. Differences between covariates existed; however, the sample size was large enough; thus, propensity score matching was feasible, and balance was achieved. I have minor remarks only:

• The (NHIS) abbreviation was mentioned in the introduction before the methods section. Thus, spelling the "National Health Interview Survey" before its abbreviation in the introduction (line 69) would be helpful.

• Some citations were missing (lines 40-41, 47, 75, 76, 78, and 79).

Kind regards,

6. PLOS authors have the option to publish the peer review history of their article (what does this mean?). If published, this will include your full peer review and any attached files.

Reviewer #1: **Yes: **yes

Reviewer #2: **Yes: **Tarek Tawfik Amin

Reviewer #3: No

---

## [Author Response · Author response to Decision Letter 0]

16 May 2023

Dear Reviewers, 

We thank you for your time and efforts to increase the quality of this manuscript. Please see our responses below. Line references refer to the changes made in the tracked changes document. 

Reviewer #1: It is no doubt that the author included a huge data and this will increase the reliability of the results but I wonder why he ignored some an important risk factors such as smoking, drinking, and hypertension. Anyhow I think this manuscript is eligible for publishing as it.

RESPONSE: We thank the reviewer for their comments, and for pointing out the absence of risk factors such as smoking, drinking and hypertension in the analysis we conducted. Unfortunately, not all important risk factors (especially behavioral risk factors) were available as variables in the data set used for this analysis, and alcohol was among those not present. Based on this comment, however, we examined the change in model fit and change in the main effects of the covid +symptoms variable when we added smoking and hypertension. We found that these additions did not appreciably improve the model fit (change in c-index of 1%) or change the effects for the primary predictor in any meaningful way. We have elected to use the simplified original model, and we have added language to the statement of limitations to make this approach (and potential limitation) clear (lines 319-324).

Reviewer #2: 

- There is a fundamental question the MS did not answered anywhere: how the authors confirmed the severity of COVID-19 severity?

RESPONSE: The survey participants were asked the following question: “How would you describe your coronavirus symptoms when they were at their worst? Would you say no symptoms, mild symptoms, moderate symptoms, or severe symptoms?” We have added this information to line 99 for additional clarification. 

- What are the traditional risk factors for chronic pain? This should be elaborated

RESPONSE: Traditional risk factors for chronic pain are outlined in lines 59-62. 

- The authors made a great effort in statistical analysis to control the possible confounders which is a novel approach for this MS.

RESPONSE: We thank the reviewer for this comment. 

Introduction:

In this section the authors did not define the term “chronic pain” and its significance.

RESPONSE: We have added the definition of chronic pain, as well as an additional reference for this definition. 

Line 44: 19% of those instead of just 19% those?

RESPONSE: We have corrected this typo. 

Methods:

- Line 81: URL for this data should be referenced.

RESPONSE: We have added this URL. 

- What do you mean by valid pain data? How did you validate the presence of chronic pain among the survey population?

RESPONSE: We have removed the word ‘valid’, as this word was intended to indicate non-missing data and data that were not coded as “refuse to respond”. 

- The classification of severity (apart from tests results) was not clear which symptoms (symptoms load) the authors identified to classify their participants accordingly?

RESPONSE: Symptom load is self-reported. Specifically, the survey question states: “How would you describe your coronavirus symptoms when they were at their worst? Would you say no symptoms, mild symptoms, moderate symptoms, or severe symptoms?” We have added this clarification to the manuscript (line 99-100).

- What are the variables used to conduct sensitivity analyses (and how many analyses were required) and what were the possible scenarios evolved?

RESPONSE: We believe the reviewer is referring to the original line 102 with this inquiry. We conducted a sensitivity analysis to ensure that asymptomatic and mildly symptomatic groups were similar in both covariate distribution and pain, before we grouped them together. Both asymptomatic and mildly symptomatic individuals reported lower prevalence of pain compared to the no covid group – indicating the findings were not being driven by either the asymptomatic or mild group. Similarly, we wanted to ensure those with moderate and severe symptoms groups had similar covariate distribution and pain before we grouped them together. The findings here were similar as well – both moderate and severe symptom groups had higher prevalence of pain compared to the no covid group, indicating these results were not being driven by just one of these symptom level groups (moderate or severe). Because these symptomatic groups behaved similarly, we had higher confidence in grouping them together. We have clarified this language in the methods section (lines 101-106).

- How the poverty income ratio was calculated? What was its significance?

RESPONSE: As per the NHIS database codebook, poverty income ratio is defined as the “ratio of family income to poverty threshold”. This is a variable that exists within the NHIS Database, however, we have added the definition to line 117 for further clarification. This variable is commonly used as a metric of socioeconomic status. It is well established that socioeconomic status is inversely related to chronic pain - see references 16 (Axon 2021) and 17(Mills 2019). 

- Multi-survey weights were applied and complex survey weights were used: what are the differences?

RESPOSNE: The term multi-level survey weights and complex survey weights are synonymous but for clarity, we have revised the term to complex survey weights throughout (line 124). 

- Line 117: remove “simple” and replace it with bivariate logistic regression.

RESPONSE: We have removed the term ‘simple’, and kept ‘Bivariate’ at the beginning of the sentence.

- I think in this section you have to mention the sensitivity analyses followed by propensity score and matching: for each you have to mention the importance and applicability to your research.

- Line 126 to line 127: this paragraph is confusing and needs more elaboration.

Results:

RESPONSE: We have added a line of clarification as to why we performed propensity score matching, specifically to reduce additional confounding that was likely occurring due to the imbalance of covariates between the three groups. (Lines 139-141). 

- How many were included in the original survey?

RESPONSE: 29,482. We have added this number to the methods, and we state/discuss related limitations in the discussion section. 

- Remove representing 135.7 million US adult citizens.

RESPONSE: After applying the complex survey weights, this is the weighted estimation of US adult citizens represented by the survey. We find it important to include this number to signify that this is a weighted survey representing the US adult population. 

- In table 1: you have to mention “the weightage” unweighted and weighted?????

- Table 1: the title of the table is not full: table should be stand alone.

- Table 1: use effective digits if possible: 79.11% (to 79.1%)

RESPONSE: Table 1 demonstrates the proportions of the frequencies as unweighted and weighted (i.e., application of the complex survey weights). This is a common practice of displaying large population -based survey data, but the terms are outlined in the methods section for clarification (lines 126-127). 

- Line 151: mention the Odds ratio and C.I for the prevalence of chronic pain among the different groups.

RESPONSE: We report the unadjusted odds ratios (CIs) in table 2. 

- Table 1: bivariate analysis should be included

RESPONSE: We include the bivariate analyses in Table 2. 

- Table 2: the same comments regarding the adjustment

RESPONSE: We include the bivariate analyses in Table 2. 

- Please revise the Odds ratio for gender (1.14 “1.04-1.24’)????

RESPONSE: We have double checked our analysis – the reported odds ratio and confidence interval are correct. 

- Figure 1A: the probability of chronic pain (not just pain is 4% higher)!!! In this figure, the dose-relationship is not working for COVID-19, as the gradient for pain is less in mild vs. COVID-19 negative, this needs explanation????

RESPONSE: We have added the word chronic prior to pain in the results (line 180). We explain a possible reason for this non-dose response relationship in the discussion (lines 235-245), but we have also added connections to prior literature (lines 246-248). 

- Line 171: COVID-19 symptoms group was found to be significantly associated with pain???? Needs elaboration: how was significantly associated???

RESPONSE: Because this is the figure legend text, we chose not to repeat information from the figure in the text of the legend. Proportions are labeled on the figure, with an asterix (and labeling) indicating the level of significance for each comparison. 

- Line 179 to line 187: this section should be placed under the statistical analysis section

RESPONSE: We believe that reporting the imbalances that were present is an important result. Further, we have added a finding that may assist in explaining the non-dose response relationship between symptoms and pain. 

Discussion section:

- Line 224: confusing “in the lowest frequency group”????

RESPONSE: We have added clarifying language here. The mild symptom group (n = 1440) has the lowest frequency within the database compared to no covid (n = 12,131) and higher symptoms (n = 1764). The final matching dataset is limited by lowest frequency group. Propensity score matching in general is a very data hungry approach, so by matching a very high proportion of this lowest frequency group (85%), we were able to preserve a great deal of the sample while still balancing the measured covariates. 

- A separate section should be dedicated to study’s limitation “most important of which is the built-in problem of cross-sectional design and scarcity of evidence to support your conclusion.

RESPONSE: Indeed, we agree that cross-sectional designs and survey data in general have limitations, and we have referred to these limitations in the discussion section (starting line 315). We also acknowledge that few studies have examined this issue, but a few smaller studies have examined de novo pain after a covid infection (lines 242-254). 

- Provide some explanation of your findings: the conflicting findings of pain in relation to mild vs. negative cases (for example).

RESPONSE: We explain a possible reason for this non-dose response relationship in the discussion (lines 235-248), and why using propensity scores to match on pre-existing risk factors for pain plays a potentially important role in interpretation of the findings.

- The conclusion section should replace that in the abstract (more realistic)

RESPONSE: The conclusion section of the manuscript and the conclusion section of the abstract are similar. We elect to maintain the abstract conclusion as is. 

Reviewer #3: 

Reviewer #3: Dear Editor and Authors,

Thank you for the opportunity to review this manuscript.

It is a cross-sectional population-based study assessing whether COVID-19 symptom severity could be a risk factor for self-reported chronic pain. Data were obtained from the National Health Interview Survey (NHIS) for the year 2021. Authors categorised the primary predictor based on testing status and symptom load into three categories: no-infection, asymptomatic/mild and moderate/severe symptoms. The primary outcome was dichotomised into never having pain and reported pain groups. A complex survey design and multi-level survey weights were used. Multivariate logistic regression analysis and 1:1:1 propensity score matching was used to account for all covariates and ensure balance and better estimates, which was feasible with the large sample size of 15,335 US individuals. A total of 3669 participants were matched.

Authors found that the prevalence of chronic pain was higher in the moderate/severe symptoms group (25.48%) than in the reference group (19.44%). Adjusted regression analysis revealed that individuals in the moderate/severe symptoms group had significantly higher odds of chronic pain (OR: 1.28, 95%CI: 1.09 to 1.51) than those with no infection. Remarkably those in the asymptomatic/mild symptoms groups had lesser odds of chronic pain (OR: 0.81, 95%CI: 0.69 to 0.96) than the reference group.

Similarly, matched adjusted regression model showed significantly higher odds of pain for the moderate/severe group than the reference group (OR: 1.45, 95%CI: 1.14 to 1.83). However, matching revealed no significant difference between the mild/ asymptomatic and the no-infection group (OR: 0.86, 95%CI: 0.68 to 1.09). Adjusted probability of chronic pain was also higher in those with moderate/severe symptoms than in the reference and the asymptomatic/mild symptoms groups, 20%, 16%, and 13%, respectively. Matched paired probabilities were also similar (22%, 16%, and 14%).

My overall impression is positive; despite the cross-sectional nature limitations, methods and analysis were rigorous, and key limitations were highlighted. Differences between covariates existed; however, the sample size was large enough; thus, propensity score matching was feasible, and balance was achieved. I have minor remarks only:My overall impression is positive; despite the cross-sectional nature limitations, methods and analysis were rigorous, and key limitations were highlighted. Differences between covariates existed; however, the sample size was large enough; thus, propensity score matching was feasible, and balance was achieved. I have minor remarks only:

• The (NHIS) abbreviation was mentioned in the introduction before the methods section. Thus, spelling the "National Health Interview Survey" before its abbreviation in the introduction (line 69) would be helpful.

RESPONSE: We thank the reviewer for their comment, and we have made this change.

• Some citations were missing (lines 40-41, 47, 75, 76, 78, and 79).

RESPONSE: We have added these citations to the manuscript. 

Again, thank you for your time. 

Sincerely, 

The Manuscript Authors

---

## [Decision Letter · Decision Letter 1]

29 May 2023

PONE-D-23-09591R1COVID-19 symptom load as a risk factor for chronic pain: a national cross-sectional studyPLOS ONE

Dear Dr. Romeiser,

Thank you for submitting your manuscript to PLOS ONE. After careful consideration, we feel that it has merit but does not fully meet PLOS ONE’s publication criteria as it currently stands. Therefore, we invite you to submit a revised version of the manuscript that addresses the points raised during the review process.

We look forward to receiving your revised manuscript.

Kind regards,

Dong Keon Yon, MD, FACAAI

Academic Editor

PLOS ONE

Journal Requirements:

**Additional Editor Comments:**

This is an excellent paper. However, the authors did not address my previous comments.

#1. IRB review was exempt for this study. -> The University IRB is exempt, but there is a CDC IRB for the National Health Interview Survey. Describe it.

#2. Ref 18 and 19 is too old (1985??). Please cite this paper. DOI: https://doi.org/10.54724/lc.2022.e18

#3. This is an excellent paper!

Reviewers' comments:

Reviewer's Responses to Questions

**Comments to the Author**

1. If the authors have adequately addressed your comments raised in a previous round of review and you feel that this manuscript is now acceptable for publication, you may indicate that here to bypass the “Comments to the Author” section, enter your conflict of interest statement in the “Confidential to Editor” section, and submit your "Accept" recommendation.

Reviewer #1: All comments have been addressed

Reviewer #2: All comments have been addressed

Reviewer #3: All comments have been addressed

2. Is the manuscript technically sound, and do the data support the conclusions?

Reviewer #1: Yes

Reviewer #2: Yes

Reviewer #3: Yes

3. Has the statistical analysis been performed appropriately and rigorously? 

Reviewer #1: Yes

Reviewer #2: Yes

Reviewer #3: Yes

4. Have the authors made all data underlying the findings in their manuscript fully available?

Reviewer #1: Yes

Reviewer #2: Yes

Reviewer #3: Yes

5. Is the manuscript presented in an intelligible fashion and written in standard English?

Reviewer #1: Yes

Reviewer #2: Yes

Reviewer #3: Yes

6. Review Comments to the Author

Reviewer #1: In regard to my comments, the author's answers convinced me.

All required questions have been answered and that all responses meet formatting specifications.

Reviewer #2: - In table 1: you mentioned % in the head rows and not need the % after the figure for each individual cell.

Reviewer #3: All comments have been addressed. Clarifications have been added, and the methods and results look clearer.

Great paper. Well done to Authors!

7. PLOS authors have the option to publish the peer review history of their article (what does this mean?). If published, this will include your full peer review and any attached files.

Reviewer #1: **Yes: **Firas Rashad Al-Samarai

Reviewer #2: **Yes: **Tarek Tawfik Amin

Reviewer #3: No

---

## [Author Response · Author response to Decision Letter 1]

6 Jun 2023

We thank the Editor and Reviewers for their comments and time. Below are our responses to their additional comments. In addition, we have reformatted our references. 

Additional Editor Comments:

This is an excellent paper. However, the authors did not address my previous comments.

RESPONSE: We thank the editor for this comment. Please know that we did not see the three comments below included in the original request for revision. 

#1. IRB review was exempt for this study. -> The University IRB is exempt, but there is a CDC IRB for the National Health Interview Survey. Describe it.

RESPONSE: We have added a line about the NCHS review board. 

#2. Ref 18 and 19 is too old (1985??). Please cite this paper. DOI: https://doi.org/10.54724/lc.2022.e18

RESPONSE: Even though the publication dates on these two manuscripts are older, these are foundational manuscripts for propensity scores. We believe these maintain their importance, and have elected to keep these foundational references. Please note that in addition, we also cite newer yet renowned manuscripts that, in tandem, provide instructional support for readers. At the editor’s request, we have added the following citation: 

Lee SW, Acharya KP. Propensity score matching for causal inference and reducing the confounding effects: statistical standard and guideline of Life Cycle Committee. Life Cycle. 2022;2:e18.

#3. This is an excellent paper!

RESPONSE: We thank the editor for their time. 

Reviewers' comments:

Reviewer #1: In regard to my comments, the author's answers convinced me.

All required questions have been answered and that all responses meet formatting specifications.

RESPONSE: We thank the reviewer for their time. 

Reviewer #2: - In table 1: you mentioned % in the head rows and not need the % after the figure for each individual cell.

RESPONSE: We have identified multiple manuscripts published within PLOS ONE that format tables with % signs included within the cells, and the column header as %. We have elected to maintain this original format.

Reviewer #3: All comments have been addressed. Clarifications have been added, and the methods and results look clearer.

Great paper. Well done to Authors!

RESPONSE: We thank the reviewer for their time.

---

## [Editor Report · Decision Letter 2]

7 Jun 2023

COVID-19 symptom load as a risk factor for chronic pain: a national cross-sectional study

PONE-D-23-09591R2

Dear Dr. Romeiser,

We’re pleased to inform you that your manuscript has been judged scientifically suitable for publication and will be formally accepted for publication once it meets all outstanding technical requirements.

Kind regards,

Dong Keon Yon, MD, FACAAI

Academic Editor

PLOS ONE

Additional Editor Comments (optional):

This is an excellent paper!
---

## [Editor Report · Acceptance letter]

13 Jun 2023

PONE-D-23-09591R2 

COVID-19 symptom load as a risk factor for chronic pain: a national cross-sectional study 

Dear Dr. Romeiser:

I'm pleased to inform you that your manuscript has been deemed suitable for publication in PLOS ONE. Congratulations! Your manuscript is now with our production department. 

Kind regards, 

on behalf of

Dr. Dong Keon Yon 

Academic Editor

PLOS ONE